# The COVID-19 pandemic and mental health in Kazakhstan

Gaukhar Mergenova[1,2] , Susan L. Rosenthal[3], Akbope Myrkassymova[2], Assel Bukharbayeva[2], Balnur Iskakova[2], Aigulsum Izekenova[2], Assel Izekenova[4], Lyailya Alekesheva[2], Maral Yerdenova[2], Kuanysh Karibayev[2], Baurzhan Zhussupov[2], Gulzhan Alimbekova[5] and Alissa Davis[6]

[1]Global Health Research Center of Central Asia, Almaty, Kazakhstan; [2]Asfendiyarov Kazakh National Medical University, Almaty, Kazakhstan; [3]Department of Psychiatry, Vagelos College of Physicians and Surgeons, Columbia University Irving Medical Center, New York, NY, USA; [4]Kenzhegali Sagadiyev University of International Business, Almaty, Kazakhstan; [5]CIOM (Public Opinion Research Centre), Almaty, Kazakhstan and [6]Columbia University School of Social Work, New York, NY, USA

**Keywords:**
anxiety; COVID-19; depression; low and middle-income countries; mental health

**Corresponding author:**
Gaukhar Mergenova;
Email: gaukhar.mergenova@gmail.com

## Abstract

The COVID-19 pandemic had significant impacts on mental health. We examined factors associated with symptoms of depression and anxiety during the COVID-19 pandemic in Kazakhstan. We surveyed 991 adults in Kazakhstan in July 2021 using multistage stratified sampling. Depression and anxiety were measured with the Patient Health Questionnaire-4. We conducted logistic regression to assess associations between depression and anxiety and socio-behavioral factors. Overall, 12.01% reported depressive symptoms and 8.38% anxiety. Higher likelihood of depression was associated with being female (AOR: 1.64; 95% CI [1.05, 2.55]), having experience with COVID-19 in the social environment (AOR: 1.85; 95% CI [1.1–3.14]), experiencing food insecurity (AOR: 1.80; 95% CI [1.11–2.89]), increased family conflict (AOR: 2.43; 95% CI [1.32–4.48]) and impaired healthcare access (AOR: 2.41; 95% CI [1.32–4.41]). Higher likelihood of anxiety was associated with being female (AOR: 3.43; 95% CI [1.91–6.15]), increased family conflict (AOR: 2.22; 95% CI [1.11–4.44]) and impaired healthcare access (AOR: 2.63; 95% CI [1.36–5.12]). Multiple factors were associated with mental health in Kazakhstan during the COVID-19 pandemic. Further research is needed to determine the extent to which these factors and their associated mental health outcomes may persist.

## Impact statement

The COVID-19 pandemic had significant impacts on mental health. Our results suggest that in Kazakhstan, women experienced higher rates of depression and anxiety than men. Rurality, limited access to healthcare services, increased family conflicts, and knowing someone who died of COVID-19 were also associated with an increased likelihood of mental health symptoms. In addition, economic vulnerability, such as food insecurity, was associated with increased depression. By identifying factors associated with greater risk, policies can be developed that either mitigate these factors (e.g., limited access to health care) or their relationship to mental health (e.g., being female or living in a rural area) so as to support the mental health of the general population of Kazakhstan.

## Introduction

The coronavirus disease 2019 (COVID-19) pandemic is a multidimensional global public health problem. Along with its effects on physical health, previous infectious disease epidemics have also had a substantial negative impact on people's mental health (Lee et al., 2007; Lau et al., 2010). The COVID-19 pandemic resulted in mass disruptions globally that impacted emotional well-being and mental health, not only due to fears around COVID-19 infection and mortality, but also due to social and behavioral factors, including strict lockdown and quarantine measures, disrupted work and school routines, and increased social isolation (Brooks et al., 2020; Campion et al., 2020; Kola et al., 2021). Particularly with regard to the latter, it has been believed that lockdowns made people feel lonely, irritable, restless and anxious (Fullana et al., 2020; Saladino et al., 2020; Gonda and Tarazi, 2022). Difficulties acquiring food and medical services, medical comorbidities and lack of specialized treatment further resulted in a substantial mental burden that in turn caused psychological distress and mental health disorders (De Sousa et al., 2020; Lai et al., 2020; Gillard et al., 2021; Rahman et al., 2021; Han et al., 2022).

Studies indicate that up to 40% of the general population experienced high levels of anxiety or distress associated with the COVID-19 pandemic and that there may be psychological and emotional trauma that would last a lifetime (Hossain et al., 2020; Jung et al., 2020; Vindegaard and Benros, 2020; Jin et al., 2021; Mauz et al., 2021; Bonati et al., 2022). Depression and anxiety disorders rank among the top debilitating medical conditions and have one of the highest socioeconomic impacts (GBD 2019 Diseases and Injuries Collaborators, 2020; GBD 2019 Mental Disorders Collaborators, 2022). Studies in both high- and low-income countries exhibit heterogeneity in factors most associated with mental health outcomes during the COVID-19 pandemic (COVID-19 Mental Disorders Collaborators, 2021; Kola et al., 2021; Shevlin et al., 2021; Bonati et al., 2022). Kazakhstan is considered an upper middle-income country but has high suicide mortality rate. According to the World Health Organization (WHO), the age-standardized suicide rate in Kazakhstan was 6.9 per 100,000) (WHO, 2021) and the UNICEF-2013 report (UNICEF, 2013) indicates that the risk of suicides among adolescents (15–19 years) in Kazakhstan is three times higher than in the Commonwealth of Independent States (CIS). Suicides are usually associated with underlying depressive conditions (Isacsson, 2000; Gotlib and Hammen, 2002).

Currently, there are limited data on the prevalence of depression and anxiety in Kazakhstan. According to official data posted on the website of the Republican Scientific and Practical Center of Mental Health, the number of registered patients with mental and behavioral disorders was 1,020.1 per 100 000 – in 2019 and 1,004.0 per 100 000 – in 2020 with depressive and anxiety disorders included in these numbers (Respublikanskiy Nauchno-Prakticheskiy Tsentr Psikhicheskogo Zdorov'ya, 2021). Furthermore, a WHO mental health report indicated that the COVID-19 pandemic negatively affected mental health globally with an increase of 28% and 26% for major depressive disorders and anxiety disorders, respectively in just one year (WHO, 2022). Important factors for mental health during the COVID-19 pandemic appear to vary based on local context, but little research has been conducted examining the impact of COVID-19 on mental health outcomes among the general population in Kazakhstan.

The first major lockdown in Kazakhstan occurred in March 2020, with severe restrictions on travel between cities, closure of entertainment and other venues, suspension of cultural events and large family and public gatherings, and strict quarantine rules. In the summer of 2021, when new vaccines were available and people were feeling some hope, the more contagious Delta variant became the predominant SARS-CoV-2 variant, leading to a dramatic increase in hospitalizations worldwide (Hart et al., 2022). In Kazakhstan, COVID-19 cases started to rise at the end of June 2021 and reached their peak in August 2021. This was also the period with the highest number of recorded daily deaths in Kazakhstan during the entire COVID-19 pandemic. As a result, Kazakhstan implemented a second lockdown in July 2021, which included reducing the operating hours of businesses and entertainment venues, prohibiting in-person dining and restricting public gatherings (UNCT, 2020; Haruna et al., 2022). These restrictions likely led to changes in health behaviors, such as physical activity, smoking and alcohol use, interpersonal relationships, such as family dynamics, and structural factors, such as income and employment and health care access, as well as increased depression and anxiety. Although there have been studies targeting specific groups and mental health in relation to or during the COVID-19 pandemic in Kazakhstan (Bolatov et al., 2020; Bazarkulova and Compton, 2021; Crape et al., 2021; Kamkhen et al., 2022; Konstantinov et al., 2022), little

is known about which specific COVID-19-related factors were associated with mental health among the general Kazakhstani population. To address this gap, we sought to examine the multi-level COVID-19 related factors associated with mental health in order to inform the country's future programmatic and policy response to this public health crisis.

## Methods

### *Study design*

We conducted a cross-sectional face-to-face survey of 1,021 participants between June 26 and July 10, 2021. Data collection was performed by the Public Opinion Research Centre. The team of Public Opinion Research Centre is an experienced team of specialists that have worked in the field for several years. We provided training sessions via Zoom for the research assistants to ensure adherence to data collection protocols, confidentiality rules and ethical principles of the study. Our team provided support and supervision to ensure high quality of the process of data collection. Once data were collected, we checked audio records and survey data, to ensure the quality of data and excluded from the final dataset data that were incomplete of low quality. Participants were recruited using a multi-stage sampling approach. Strata were identified in the first stage, which represented the administrative regions of the country, separated into urban and rural populations. The number of respondents in each stratum corresponded to the population living there. At the second stage, the settlements where the survey would be conducted were chosen: the region's largest city and a randomly selected rural settlement. A random route sample was used to determine households in the third stage. Streets were chosen at random from a list of streets to generate random routes throughout the urban and rural settlements. The starting point of the route was chosen randomly by picking a house number on the designated street. Then, in increasing order, every third house was selected. If an apartment complex was picked for the survey, a systematic sample was employed to identify every fifth apartment in the building. In households, interviewers recruited participants applying a gender and age frequency-match approach. General population data were obtained from official 2019 census (Bureau of National Statistics, 2020). Oral informed consent was obtained from all participants of the study before the start of the survey. All databases, folder and personal computers were password-protected. All databases were deidentified prior to the start of data cleaning and analysis. File linking identifiable information and ID numbers of participants were only available for the limited number of research assistants who were involved in data collection and the principal investigator. The average time required to complete the survey was 40–60 minutes. Out of 1,021 respondents, 30 were dropped due to incomplete surveys. The final sample consisted of 991 adults.

### *Measures*

#### *Dependent variables:* Depression and anxiety

To measure the presence of depressive and anxiety symptoms, we used the Patient Health Questionnaire-4 (PHQ-4). The PHQ-4 is an ultra-brief tool for detecting both anxiety and depressive disorders (Kroenke et al., 2009). It has been used in numerous studies in several countries (Schnell and Krampe, 2020; Zhang et al., 2020a; Daly and Robinson, 2021; Workneh et al., 2021). An elevated PHQ-4 score is not diagnostic, but is an indicator for further

inquiry to establish the presence or absence of a clinical disorder warranting treatment. The PHQ-4 begins with the stem question: "Over the last 2 weeks, how often have you been bothered by the following problems?" Responses are scored as 0 ("not at all"), 1 ("several days"), 2 ("more than half the days") or 3 ("nearly every day"). The total composite score of PHQ-4 ranges from 0 to 12, and goes from normal (0–2) to mild (3–5) to moderate (6–8) to severe (9–12) (Cronbach Alpha = 0.76). Positive screening for anxiety was defined as a score of ≥3 on the General Anxiety Disorder (GAD)-2 (which assesses "feeling nervous, anxious or on edge" and "not being able to stop worrying") of the PHQ-4 (Cronbach Alpha = 0.67) (Kroenke et al., 2007; Levis et al., 2020), and positive screening for depression was defined as a score of ≥3 on the 2-item Depression subscale (PHQ-2) which assesses "feeling down, depressed and hopeless" and "little interest or pleasure in doing things")) of the PHQ-4 (Cronbach Alpha = 0.61) (Kroenke et al., 2003; Löwe et al., 2005; Bisby et al., 2022). PHQ-4 is a subset of the Patient Health Questionnaire (PHQ-9), which had been previously validated in Russian. In Kazakhstan, historically, population is fluent in Russian (Pogosova et al., 2014).

### Independent variables

*Sociodemographic characteristics* included self-reported age, gender, education, type of residence, employment status, and if they had adults older than 65 in their households.

*COVID-19 related experiences and behavior.* Participants self-reported if they thought they ever had a COVID-19 infection (yes/no). They were also asked if they knew someone who was infected with COVID-19 or had died of COVID-19 and were classified into three categories (knew someone who had died/knew someone infected, but did not die /did not know anyone who had died or was infected).

Likelihood of severe COVID-19 was assessed with 5-point Likert-type questions: "In your opinion, how severe would contracting COVID-19 be for you?" (1 – "very mild" to 5 – "very severe") (Brewer et al., 2007).

We also asked about changes in terms of level of conflicts in the home at the time of COVID-19 pandemic using the question "During the COVID-19 pandemic, have the level of conflicts in your home" with dichotomized categories of response options: decreased or stayed the same compared to the period before the COVID-19 pandemic and Increased compared to the period before the COVID-19 pandemic.

Participants self-reported changes regarding their health behaviors, including smoking, alcohol use and physical activity. For example, "How has your physical activity level changed during the pandemic (i.e., from March 2020 to the present) compared to the period before the COVID-19 pandemic?" with response options: "has not changed," "decreased" and "increased." We then used dichotomized variables (decreased/has not changed vs. increased).

*Economic vulnerabilities and healthcare service access.* We asked participants questions about their change of financial status (Deteriorated/Has not changed/Improved/Do not know) and working conditions (Deteriorated/Has not changed/Improved) during the pandemic. We also asked participants if they faced food insecurity (yes/no) during the pandemic.

To evaluate how changes regarding work might affect mental health, we asked if working conditions worsened (yes/no).

We evaluated changes in healthcare access and asked participants if their medical care for other non-COVID-19 illnesses changed during the COVID-19 pandemic compared to the pre-

pandemic period. Responses were dichotomized for analysis: did not have problems with healthcare access (No, I have not had to use other healthcare services during the pandemic/No, my healthcare remains the same as before the pandemic/Yes – I have been offered remote appointments via telephone or video call); and had problems with healthcare access (Yes – I have had appointments and procedures postponed or canceled/Yes – I have been unable to make appointments for new health issues).

### Statistical analysis

Participant characteristics were described using means and standard deviations (SDs) for continuous variables and frequencies and percentages for categorical variables. The PHQ-4 score was categorized according to questions measuring depression and anxiety, indicating the presence or absence of depression (PHQ-2 ≥ 3) or anxiety symptoms (GAD-2 ≥ 3) (Kroenke et al., 2003, 2007).

To examine which multi-level factors were associated with mental health symptoms, we conducted logistic regression analyses. First, we conducted bivariate analyses to identify potential associations with all multi-level factors we hypothesized would be associated with mental health symptoms. Then all variables that were significant for depression symptoms or anxiety symptoms at the $p \leq 0.10$ level and were entered simultaneously into a multivariable logistic regression model (Heinze et al., 2018). For the final multivariable model, we used a significance level of $p \leq 0.05$. We checked variables for multicollinearity before including them into the model. We used SAS 9.4 for analysis.

### Ethical approvals.

The study was approved by the ethical committee No. 10 of the Asfendiyarov Kazakh National Medical University on September 30, 2020.

### Role of the funding source

Funded by the Science Committee of the Ministry of Science and Higher Education of the Republic of Kazakhstan № AP09260497 "The Impact of the COVID-19 Pandemic and Restrictive Measures on Lifestyles and Access to Health Care in Kazakhstan." The funders had no role in study design; data collection, analysis, interpretation; writing; or the decision to submit the article.

## Results

### Sample characteristics

Table 1 summarizes the sociodemographic characteristics of the study population and the variables we used in our analysis. The mean age of participants was 41.1 (SD = 15.00) years old and about half of the sample were women (n = 524, 52.9%). The majority of participants were married (n = 618, 62.4%), lived in urban areas (n = 591, 59.6%), and were employed full-time (n = 529, 53.4%) or part-time (n = 94 (9.49%)). Over a third of participants had a postgraduate degree (completed a bachelor or higher degree) (n = 412, 41.6%). Less than a fifth (n = 162, 16.4%) lived with a person who was older than 65 years old. Over a third experienced deterioration in their financial status (n = 362, 36.5%) and more than third (n = 410, 41.4%) reported food insecurity during the pandemic.

**Table 1.** Sociodemographic characteristics and mental health of the sample (*n* = 991)

| Characteristics | Total sample, *n* (%) |
| --- | --- |
| | Mean (SD) |
| Age | 41.1 (15.0) |
| | *n* (%) |
| Gender | |
| Male | 467 (47.12) |
| Female | 524 (52.88) |
| Marital status | |
| Married, in relationships | 618 (62.36) |
| Single, widowed, divorced | 373 (37.64) |
| Education | |
| High and postgraduate | 412 (41.57) |
| Up to secondary | 242 (24.42) |
| Specialized secondary | 337 (34.01) |
| Current employment status | |
| Full-time | 529 (53.38) |
| Part-time | 94 (9.49) |
| Unemployed | 80 (8.07) |
| Other | 288 (29.06) |
| Area of residence | |
| Rural | 400 (40.36) |
| Urban | 591 (59.64) |
| Living with older people (65+) | 162 (16.35) |
| Perception of COVID-19 severity (mean, SD) | 2.55 (0.87) |
| COVID-19 self-report or diagnose | 180 (18.16) |
| Knowing someone with COVID-19 | |
| Knows someone who died of COVID-19 | 192 (19.37) |
| Knows someone who had COVID-19 | 191 (19.27) |
| Does not know anyone with COVID-19 | 608 (61.35) |
| Physical activity decreased | 197 (19.88) |
| Alcohol consumption increased | 38 (3.83) |
| Smoking increased | 17 (1.72) |
| Conflicts increased | 73 (7.37) |
| Financial status deteriorated | 362 (36.53) |
| Food insecurity | 410 (41.37) |
| Worked remotely from home | 76 (7.67) |
| Working conditions worsened | 89 (8.98) |
| Had problems with healthcare access | 78 (7.87) |
| Mental health symptoms (PHQ-4) | |
| Anxiety symptoms (GAD-2 ≥ 3) | 83 (8.38) |
| Depression symptoms (PHQ-2 ≥ 3) | 119 (12.01) |
| Moderate (6 ≤ PHQ-4 < 9) | 44 (4.84) |
| Severe (PHQ-4 > 9) | 19 (2.12) |

SD, standard deviation.

### Symptoms of anxiety and depression

The mean value of the PHQ-4 score was 1.6 (SD: 2.26). Nearly a fifth of respondents *n* = 190 (19.17%) reported at least mild mental health symptoms, 4.8% (*n* = 48) had moderate symptoms and 2.1% had severe symptoms (Figure 1). In the total sample, 12.0% of participants had positive screening for depression (≥3 PHQ-2) and 8.4% of participants had positive screening for anxiety (≥3 GAD-2).

### COVID-19-related experiences and health behavior

About one-fifth of our respondents think that they had COVID-19 at least one time in their life (*n* = 180, 18.2%). Two-thirds of the sample did not know anyone who was infected with COVID-19 (*n* = 608, 61.4%).

In terms of adverse behavioral changes, almost a fifth (19.9%) of the sample reported decreased physical activity (*n* = 197). 7.4% reported increased family conflicts level (*n* = 73) and 7.9% reported problems accessing healthcare (*n* = 78). A minority (3.8%) reported increased alcohol consumption (*n* = 38) and 1.7% reported increased smoking (*n* = 17).

In the multivariable regression analysis, regarding depressions symptoms we found that being female (AOR: 1.64; 95% CI [1.05, 2.55]), living in a rural area (AOR: 1.75; 95% CI [1.14–2.68]), perceiving greater severity of a COVID-19 infection (AOR: 1.28; 95% CI [1.00, 1.63), knowing someone who died from COVID-19 (AOR: 1.85; 95% CI [1.1–3.14]) or someone who was infected with COVID-19 (AOR: 1.90; 95% CI [1.12–3.17]), having increased conflict in the home (AOR: 2.43; 95% CI [1.32–4.48]), having food insecurity (AOR: 1.80; 95% CI [1.11–2.90]) or having problems accessing health care (AOR: 2.41; 95% CI [1.32–4.41]) were associated with higher odds of having depressive symptoms (Table 2).

For anxiety, we found that being female (AOR: 3.43; 95% CI [1.91–6.15]), having decreased physical activity (AOR: 2.11; 95% CI [1.24–3.57]), perceiving greater severity of a COVID-19 infection (AOR: 1.45; 95% CI [1.09–1.92]), having increased conflict in the home (AOR: 2.22; 95% CI [1.11–4.44]), and having problems accessing healthcare (AOR: 2.63; 95% CI [1.36–5.12]) were associated with higher odds of having anxiety symptoms in the multivariable model (Table 2).

### Discussion

Consistently with other studies, we found that multiple factors associated with depression and anxiety symptoms among the general population of Kazakhstan during the COVID-19 pandemic, including gender, home, economic, work and healthcare factors. Although numerous studies have shown that the COVID-19 pandemic had an adverse impact on mental health with increases in depression and anxiety, the results of these studies are highly heterogeneous, suggesting each country has a unique combination of different factors that may be affecting the mental health of their population (Vindegaard and Benros, 2020; Jin et al., 2021). To our knowledge, our study is the first study to assess factors associated with mental health during the COVID-19 pandemic variant Delta wave among the general population in Kazakhstan and fills an important gap in the literature.

Our findings are consistent with the literature in regards to women experiencing higher rates of depression and anxiety during the pandemic (Hou et al., 2020; Jung et al., 2020; Xiong et al., 2020). Many research documented that women usually have more anxiety

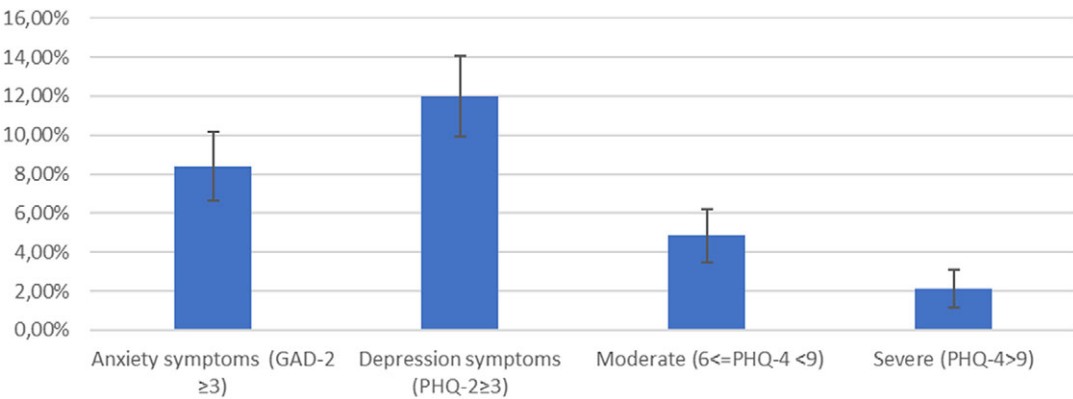

**Figure 1.** Population proportion with PHQ-4, PHQ-2, GAD-2 elevated scores.

mood disorder than men (Pigott, 1999; Kuehner, 2003; Seedat et al., 2009). Some studies suggest that public health measures such as lockdown worsened the pre-existing issues of vulnerable groups, including women (Kola et al., 2021). This could also be explained by a number of different factors that we were unable to assess such as biological and social factors especially in countries with high levels of gender inequality (Urbaeva, 2019; Oginni et al., 2021; Turusbekova et al., 2022). A systematic review conducted on 32 studies from across the globe suggested high rates of domestic violence and abuse against women during the pandemic that may have led to psychiatric distress (Kourti et al., 2023). In less extreme situations, women may experience an increased burden of household chores during lockdown due to traditional roles and imbalanced distribution of household responsibilities between women and men.

Consistent with other studies, decreased physical activity was associated with increased depression and anxiety (Stanton et al., 2020; Violant-Holz et al., 2020; Zhang et al., 2020b). However, we cannot conclude the direction of these associations (Rebar et al., 2015; Lesser and Nienhuis, 2020; Meyer et al., 2020). In a cross-sectional study of 3,052 U.S. adults, individuals who decreased physical activity had stronger/higher depressive symptoms and stress compared to those who maintained adherence to physical activity. In another multi-country cross-sectional study of physical activity and mental health among adults during the initial phases of the COVID-19 pandemic, participants who reported decreases in exercise behavior had worse mental health compared to those who had an increase or no change in their exercise behavior (Faulkner et al., 2021).

Those respondents who reported increased conflicts in family had higher odds of having symptoms of both depression and anxiety. The study among young adults aimed to understand the role of family conflict in young adult well-being found that people from families experiencing higher than usual levels of family conflict experienced more anxiety (Wang et al., 2022). In a cross-sectional study conducted by Kuśnierz et al. (2022), it was suggested that work–family conflicts and family–work conflicts are related to the worsening of mental health, including high symptoms of stress, anxiety and depression, and decreased physical health and life satisfaction.

We found that knowing someone with COVID-19 or who died from COVID-19 was associated with higher odds of reporting depressive symptoms. It is possible that those who have heard someone with severe COVID-19 infection symptoms with lethal outcomes can be more fearful of the infection and its severity. Moreover, the pandemic has changed the regular grieving process

for the deceased due to the restrictions on funeral rituals and might have led to an increased anxiety and anger among the loved ones of the deceased. The fact that many deaths from COVID-19 during the pandemic occurred at the medical facilities in isolation may have worsened this situation due to the lost chance of saying farewells (Mortazavi et al., 2021). Both qualitative and quantitative literature on the subject are consistent on the psychological burden of death from COVID-19 on psychological well-being of the relatives and friends of the deceased (Das et al., 2021; Mayland et al., 2021; Mohammadi et al., 2021; Mortazavi et al., 2021; Aguiar et al., 2022; Hernández-Fernández and Meneses-Falcón, 2022).

Living in a rural area was associated with higher odds of having depressive symptoms, which is in contrast to some other studies in China in which anxiety and depression were higher among urban residents (Zhang et al., 2021). Higher rates of depression and anxiety among rural residents have important implications, as rural areas generally tend to have poorer access to health services, particularly for mental health (Fitzmaurice, 2021; Tulegenova et al., 2022). Moreover, the ancillary effects of efforts to contain the pandemic, including lockdown, closure of schools and reallocation of health resources can be especially long-lasting and devastating to poor and vulnerable people in countries with weak social protection systems and insufficient economic resources (Kola et al., 2021).

Factors of economic vulnerability, like food insecurity, were associated with worsening of mental health and reporting symptoms of depression. A bidirectional association between food insecurity and mental health has been well described prior to the pandemic (Maynard et al., 2018). A global analysis of nationally representative surveys conducted in 149 countries found that food insecurity has a dose-response relationship with poor mental health status independent of socioeconomic and demographic characteristics (Jones, 2017). Stress levels, a potential contributor to poor mental health, have been found to increase as food insecurity deteriorates (Rahman et al., 2021). Other studies have found that the number of households experiencing food insecurity has increased during the COVID-19 pandemic (Lim et al., 2022). Moreover, food-insecure subjects were more likely to have an abnormal mental health screen compared to food-secure subjects (Lim et al., 2022).

Next, we found a strong association of impaired access to healthcare services with symptoms of depression and anxiety during the pandemic. Feeling anxious about COVID-19 or depressive feelings may serve as a barrier to reach the medical care needed, while canceled appointments and restricted access to healthcare can

**Table 2.** Bivariate and adjusted logistic regression estimates of odds ratios and 95% confidence intervals for association between depression and anxiety and studied variables

| Categorical variables | Frequency | Depression symptoms (PHQ-2 ≥ 3) | | Anxiety symptoms (GAD-2 ≥ 3) | |
|---|---|---|---|---|---|
| | | Bivariate unadjusted OR [95% CI] | Multivariable adjusted OR [95% CI] | Bivariate unadjusted OR [95% CI] | Multivariable adjusted OR [95% CI] |
| Age | 41.1 (15.0) | 1.00 (0.99, 1.01) | 1.00 (0.98, 1.01) | 1.00 (0.99, 1.02) | 1.00 (0.99, 1.02) |
| Gender | | | | | |
| Male | 467 (47.1) | ref | ref | ref | ref |
| Female | 524 (52.9) | 1.82 (1.22, 2.71)*** | 1.64 (1.05, 2.55)* | 3.53 (2.06, 6.05)*** | 3.43 (1.91, 6.15)*** |
| Education | | | | | |
| Completed high or postgraduate degree | 412 (41.6) | ref | ref | ref | ref |
| Up to secondary | 242 (24.4) | 1.11 (0.70, 1.76) | 1.22 (0.72, 2.07) | 1.06 (0.60, 1.87) | 1.07 (0.57, 2.03) |
| Specialized secondary | 337 (34.0) | 0.71 (0.45, 1.13) | 0.67 (0.41, 1.12) | 1.01 (0.60, 1.70) | 0.85 (0.48, 1.51) |
| Current employment status | | | | | |
| Full-time | 529 (53.4) | ref | ref | ref | ref |
| Part-time | 94 (9.5) | 1.51 (0.82, 2.80) | 1.32 (0.68, 2.59) | 2.47 (1.27, 4.80)** | 1.94 (0.93, 4.02) |
| Unemployed | 80 (8.1) | 1.14 (0.56, 2.33) | 1.19 (0.55, 2.58) | 0.94 (0.36, 2.48) | 0.95 (0.34, 2.66) |
| Other | 288 (29.1) | 1.10 (0.71,1.72) | 0.99 (0.60, 1.63) | 1.58 (0.95, 2.64) | 1.10 (0.62, 1.96) |
| Area of residence | | | | | |
| Rural | 400 (40.4) | 1.42 (0.97, 2.08) | 1.75 (1.14, 2.68)* | 1.27 (0.81, 2.00) | 1.42 (0.86, 2.35) |
| Urban | 591 (59.6) | ref | ref | ref | ref |
| Living with older people (65+) | 162 (16.4) | 1.51 (0.94, 2.43) | 1.32 (0.79, 2.21) | 1.71 (1.00, 2.92)* | 1.34 (0.74, 2.41) |
| COVID-19 self-report or diagnose | 180 (18.2) | 1.89 (1.22, 2.94)*** | 1.30(0.79, 2.15) | 1.38 (0.80, 2.36) | 1.04 (0.56, 1.94) |
| Perception of COVID-19 severity (mean, SD) | 2.55 (0.9) | 1.46 (1.17, 1.81)*** | 1.28 (1.00, 1.63)* | 1.61 (1.25, 2.07)*** | 1.45 (1.09, 1.92)* |
| Knowing someone with COVID-19 | | | | | |
| Does not know anyone with COVID-19 | 608 (61.4) | ref | ref | ref | ref |
| Knows someone with COVID-19 | 191 (19.3) | 1.99 (1.24, 3.20)*** | 1.89 (1.12, 3.17)* | 1.26 (0.72, 2.20) | 1.23 (0.66, 2.27) |
| Knows someone who died from COVID-19 | 192 (19.4) | 2.21 (1.39, 3.51)*** | 1.85 (1.10, 3.14)* | 0.97 (0.53, 1.77) | 0.69 (0.35, 1.37) |
| Physical activity decreased | 197 (19.9) | 2.01 (1.31, 3.07)*** | 1.48 (0.93, 2.36) | 2.51 (1.56, 4.05)*** | 2.11 (1.24, 3.57)** |
| Alcohol consumption increased | 38(3.8) | 2.77 (1.31, 5.85)** | 1.78 (0.75, 4.20) | 2.13 (0.87, 5.26) | 1.64 (0.60, 4.47) |
| Smoking increased | 17 (1.7) | 3.14 (1.09, 9.09)* | 1.73 (0.50, 6.01) | 2.40 (0.67, 8.51) | 1.48 (0.32, 6.79) |
| Conflicts increased | 73 (7.4) | 3.38 (1.95, 5.85)*** | 2.43 (1.32, 4.48)*** | 3.23 (1.74, 6.00)*** | 2.22 (1.11, 4.44)* |
| Financial status deteriorated | 362 (36.5) | 1.52 (1.03, 2.24)* | 0.98 (0.61, 1.59) | 1.52 (0.97, 2.39) | 1.0 (0.56, 1.76) |
| Had food insecurity | 410 (41.4) | 2.24 (1.51, 3.30)*** | 1.80 (1.11, 2.89)* | 1.96 (1.25, 3.09)*** | 1.32 (0.75, 2.32) |
| Healthcare access problems | 78 (7.9) | 2.84 (1.64, 4.91)*** | 2.41 (1.32, 4.41)*** | 3.26 (1.78, 5.96)*** | 2.63 (1.36, 5.12)*** |
| Working conditions worsened | 89 (9.0) | 1.42 (0.77, 2.60) | 1.01 (0.50, 2.03) | 2.03 (1.08, 3.84)* | 1.74 (0.82, 3.69) |

*$p < 0.05$;
**$p < 0.01$;
***$p < 0.005$.

also deteriorate one's mental well-being. Furthermore, some studies suggest that quarantine measures combined with restrictions in getting physical medical appointments may have exacerbated the existing mental health difficulties during the pandemic (Gillard, et al., 2021; Kola et al., 2021). Countries with fragile healthcare systems and scarce sources struggled the most to provide equal access to adequate medical interventions (De Sousa et al., 2020; Vigo et al., 2020). This can also be relatable to Kazakhstani healthcare settings due to extreme shortages in healthcare capacity including personnel, equipment and medication supply during the

pandemic (Haruna et al., 2022). Consequently, failed access to care needed may have added an extra burden and anxiety among the general public on top of the existing concerns over the fear of infection, financial and psychological difficulties.

There is a scarcity of scientific literature on the impact of limited access to healthcare on one's psychological health during the pandemic in Kazakhstan. Nevertheless, the available sources suggest that Kazakhstan had a number of challenges prior to COVID-19 such as shortage in healthcare funding, high prevalence of chronic diseases, and limited access to medical care (Haruna et al., 2022). The

emergence of the pandemic compounded the existing issues leading to an acute shortage of essential medicines and lack of hospital beds. For example, data from 2016 indicate an availability of 4.8 beds/1,000 people and a healthcare workforce of about 252,000, among them 74,600 were medical doctors. However, this coverage varied greatly across rural (61 physicians per 10,000 population in urban areas compared to 15 physicians per 10,000 in remote areas) and urban residencies. Considering such statistics, many other countries with the similar income levels (e.g., Hungary, Poland, Turkey) outperformed Kazakhstan in terms of access to medical care prior to COVID-19. One source suggests that the depressed salary among the healthcare staff to be one of the driving reasons for having low numbers of medical personnel in the country (Kumenov, 2021). Furthermore, the attempt taken by the local government on paying extra salaries to the medical staff engaged in COVID-19 care had little impact on access to care. These matters should be addressed more in future studies due to their negative impact on health outcomes.

### Strengths and limitations

The current study has several strengths and limitations. This is the first large-scale study that we are aware of that examined the population's mental health and its association with other characteristics in Kazakhstan during the peak of the COVID-19 pandemic. We used a multi-stage stratification sampling approach to increase the generalizability of findings. However, the likelihood of systematic selection bias influencing the accuracy of the estimations cannot be ruled out.

We are unable to determine the effects of COVID-19 or other factors on mental health due to the study's cross-sectional design, as longitudinal or experimental research is needed to investigate cause-effect relationships. However, our study enabled us to examine how multi-level stressors and socioeconomic factors affected the mental health of a general population sample during the height of the COVID-19 pandemic, when COVID-19 cases were spreading rapidly in Kazakhstan and restrictive measures were being imposed in all regions of the country. While many studies sampled general populations during the early stages of the pandemic (March–April 2020), to our knowledge, our study is one of the few studies to investigate factors associated with depression and anxiety symptoms during the second wave of the pandemic caused by Delta variant in Kazakhstan.

### Conclusions

We found a number of individual-, interpersonal- and structural-level factors associated with mental health symptoms among a sample of the general population in Kazakhstan during the second wave of COVID-19 pandemic.

Our data suggests that individuals living in rural areas had disproportionately high levels of mental health symptoms. People living in rural areas can be especially vulnerable in times of crisis because of insufficient infrastructure and inadequate access to social services including health care.

Strong associations found between economic vulnerability factors and mental health symptoms are concerning, since those factors may persist as a result of the prolonged negative impact of the COVID-19 pandemic on the economy of Kazakhstan.

Self-reported increased conflict in the home was associated with mental health symptoms, yet it is unclear whether conflict levels in the home have decreased since the height of the pandemic.

Many factors that are associated with adverse mental health outcomes have existed before the pandemic, but pandemic may have exacerbated these factors increasing negative impacted mental health of people. Given that many adverse impacts of the COVID-19 pandemic continue to persist (e.g., economic problems, prolonged illness, shortages of healthcare workers in rural areas), our study supports the need for policy responses that are focused on mitigating of influence of these factors on mental health of the population of Kazakhstan. Further research is needed to determine the extent to which these factors and their associated mental health outcomes may persist. It is important to continue to monitor the mental health of populations longitudinally in order to prevent long-term unfavorable mental health outcomes.

Special attention should be focused on healthcare access in rural areas at the times of crisis in the future. Social care protection, programs to support families disproportionately impacted by COVID-19 should be considered as important part of response policy at the time of crisis to minimize negative consequences on population health and well-being in Kazakhstan.

**Open peer review.** To view the open peer review materials for this article, please visit http://doi.org/10.1017/gmh.2023.46.

**Data availability statement.** According to ethics committee requirements and informed consent, data cannot be shared publicly. Data syntax will be shared upon request to the corresponding author.

**Acknowledgments.** We would like to acknowledge the following persons and institutes who contributed to this study: the study participants for dedicating time to respond to the surveys, the Public Opinion Research Centre for collecting data and the Asfendiyarov Kazakh National Medical University and the New York State International Training and Research Program (D43 TW010046) for trainings in epidemiology and biostatistics of our research team members. We also appreciate the efforts of anonymous reviewers for providing constructive feedback to our manuscript in revision rounds enabling us to make the final publication form.

**Author contribution.** G.M., A.M., A.B. and B.I. collaborated on drafting the manuscript. S.L.R., A.D. and G.M. contributed to revisions of manuscript drafts. G.M. contributed to data analysis and outlining the research objectives. B.Z. contributed to sampling design and data analysis. L.A., M.Y., K.K., G.A., Ai.I. and As.I. contributed substantially to the conception and design of the work. All authors had final approval of the version to be published and agreed to be accountable for all aspects of the work.

**Financial support.** This research has been funded by the Science Committee of the Ministry of Science and Higher Education of the Republic of Kazakhstan (Grant Number AP09260497).

**Competing interest.** The authors declare no conflicts of interest.

**Ethics statement.** The authors assert that all procedures contributing to this work comply with the ethical standards of the relevant national and institutional committees on human experimentation and with the Helsinki Declaration of 1975, as revised in 2008.

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
