## [Reviewer Report]

Dear Prof Judy Bass and Dr Dixon Chibanda, 

We would like to submit our manuscript entitled “The Covid-19 pandemic and mental health in Kazakhstan” as an original research to the Global Mental Health Journal. 

Reports on potential impact of COVID-19 on mental health have been increasing since the start of the pandemic. However, little evidence is available from Kazakhstani setting about this issue which can also be explained by the fact that the pandemic is relatively new among human populations. In the current study we attempted to investigate the factors associated with depression and anxiety during the pandemic of COVID-19 in Kazakhstan. 

Our team of young researchers from Kazakhstan conducted cross-sectional face-to-face surveys in Kazakhstan during the month of July in 2021. We hypothesized that COVID-19 related experiences, health behavior during the pandemic, economic and structural changes may affect mental health of people in Kazakhstan.

As the results of our study demonstrate, economic vulnerability, food insecurity, impaired access to care may play a lead to higher rates of depression and anxiety during the public health crisis such as COVID-19 pandemic. These findings have great implications on the gaps in medical care provision during the crisis times and economic insecurity of the population. This study may also suggest more investigation into the associations between multilevel factors and mental health.

This is the first national-wide study addressing the issue of mental health and COVID-19 in our understanding that may serve as a source for future investigations and interventions on the topic. We also confirm that this study has never been published (neither under consideration) and that all authors agree to submitting the current manuscript to.

The grant was supported by the Science Committee of the Ministry of Science and Higher Education of the Republic of Kazakhstan (Grant Number AP09260497)

Please address all correspondence to gaukhar.mergenova@gmail.com

We look forward to hearing from you at your earliest convenience.

Yours sincerely,

Gaukhar Mergenova

---

## [Reviewer Report]

This manuscript describes a study aimed to examine factors associated with symptoms of depression and anxiety during the COVID-19 pandemic in Kazakhstan. In general, the study is interesting, relevant, and undoubtedly of scientific and practical significance. However, there are the following notes:

1. Why was the significance level set to p<=0.10?

2. Indicate the date of the protocol of the local ethical committee.

---

## [Reviewer Report]

While the Introduction section provides a good overview of the general impact of COVID-19 on mental health as this is the aim of the paper, but it would be helpful to provide more specific information about the mental health challenges faced by the general population in Kazakhstan within the broader context. This information could include statistics on prevalence rates, specific mental health disorders, or relevant socioeconomic factors.

In lines 70 and 90, suggest that the authors avoid phrases like “it has been believed” or “little research has been conducted” without providing specific references or evidence to support these claims.

For the Methods section, the use of a multi-stage sampling approach to recruit participants is appropriate for obtaining a representative sample. However, additional details regarding the selection process, such as the specific criteria used for choosing the settlements and households, would enhance the clarity of the sampling method.

In lines 98-99, the authors mention that the data collection was performed by the Public Opinion Research Centre, but it would be beneficial to provide more information about the training and supervision of the interviewers. Including details about any quality control measures taken during data collection would also strengthen the methodological rigor of the study.

Within this section, the authors adequately describe the measures used to assess depressive and anxiety symptoms (PHQ-4), as well as the cutoff scores for defining positive screening for depression and anxiety. However, it would be helpful to provide references for the validation of these measures in the Kazakhstani population or similar cultural contexts.

In lines 179-180, the authors highlight the ethical approval process, however, it would be useful to specify any informed consent procedures or confidentiality measures implemented during the study as well in this section.

In the Results section, the authors document information on participants' experiences related to COVID-19, such as self-reported COVID-19 infection, knowledge of others infected with COVID-19, and changes in health behaviors and this data helps to assess the impact of the pandemic on the study population. The authors also summarized the results of the multivariable regression analysis, highlighting the factors associated with depressive and anxiety symptoms. The odds ratios (AOR) and corresponding confidence intervals (CI) are provided, along with the variables found to be statistically significant. This information allows readers to understand the relationships between independent variables and mental health outcomes.

In the Discussion section, the authors provide a comprehensive analysis of the factors associated with depression and anxiety symptoms among the general population in Kazakhstan during the COVID-19 pandemic. It highlights the unique combination of factors affecting mental health in Kazakhstan and emphasizes the importance of understanding country-specific contexts when studying the impact of the pandemic on mental health.

---

## [Reviewer Report]

Dear Prof Judy Bass and Dr Dixon Chibanda, 

We would like to submit our manuscript entitled “The Covid-19 pandemic and mental health in Kazakhstan” as an original research to the Global Mental Health Journal. 

Reports on potential impact of COVID-19 on mental health have been increasing since the start of the pandemic. However, little evidence is available from Kazakhstani setting about this issue which can also be explained by the fact that the pandemic is relatively new among human populations. In the current study we attempted to investigate the factors associated with depression and anxiety during the pandemic of COVID-19 in Kazakhstan. 

Our team of young researchers from Kazakhstan conducted cross-sectional face-to-face surveys in Kazakhstan during the month of July in 2021. We hypothesized that COVID-19 related experiences, health behavior during the pandemic, economic and structural changes may affect mental health of people in Kazakhstan.

As the results of our study demonstrate, economic vulnerability, food insecurity, impaired access to care may play a lead to higher rates of depression and anxiety during the public health crisis such as COVID-19 pandemic. These findings have great implications on the gaps in medical care provision during the crisis times and economic insecurity of the population. This study may also suggest more investigation into the associations between multilevel factors and mental health.

This is the first national-wide study addressing the issue of mental health and COVID-19 in our understanding that may serve as a source for future investigations and interventions on the topic. We also confirm that this study has never been published (neither under consideration) and that all authors agree to submitting the current manuscript. 

The grant was supported by the Science Committee of the Ministry of Science and Higher Education of the Republic of Kazakhstan (Grant Number AP09260497)

Please address all correspondence to gaukhar.mergenova@gmail.com

We look forward to hearing from you at your earliest convenience.

Yours sincerely,

Gaukhar Mergenova